# Strategies to Assess the Impact of Sustainable Functional Food Ingredients on Gut Microbiota

**DOI:** 10.3390/foods12112209

**Published:** 2023-05-31

**Authors:** Nelson Mota de Carvalho, Diana Luazi Oliveira, Célia Maria Costa, Manuela Estevez Pintado, Ana Raquel Madureira

**Affiliations:** 1CBQF—Centro de Biotecnologia e Química Fina—Laboratório Associado, Escola Superior de Biotecnologia, Universidade Católica Portuguesa, Rua Diogo Botelho, 1327, 4169-005 Porto, Portugal; ncarvalho@ucp.pt (N.M.d.C.); cfcosta@ucp.pt (C.M.C.); mpintado@ucp.pt (M.E.P.); 2Research and Innovation Unit—Instituto de Investigação e Inovação em Saúde, Universidade do Porto, Rua Alfredo Allen, 208, 4200-135 Porto, Portugal; dianao@i3s.up.pt

**Keywords:** food matrices, functional ingredients, supplementation, gut microbiota, metagenomic, metabolomics, *in vitro* models, clinical trial, circular economy

## Abstract

Nowadays, it is evident that food ingredients have different roles and distinct health benefits to the consumer. Over the past years, the interest in functional foods, especially those targeting gut health, has grown significantly. The use of industrial byproducts as a source of new functional and sustainable ingredients as a response to such demands has raised interest. However, the properties of these ingredients can be affected once incorporated into different food matrices. Therefore, when searching for the least costly and most suitable, beneficial, and sustainable formulations, it is necessary to understand how such ingredients perform when supplemented in different food matrices and how they impact the host’s health. As proposed in this manuscript, the ingredients’ properties can be first evaluated using *in vitro* gastrointestinal tract (GIT) simulation models prior to validation through human clinical trials. *In vitro* models are powerful tools that mimic the physicochemical and physiological conditions of the GIT, enabling prediction of the potentials of functional ingredients per se and when incorporated into a food matrix. Understanding how newly developed ingredients from undervalued agro-industrial sources behave as supplements supports the development of new and more sustainable functional foods while scientifically backing up health-benefits claims.

## 1. Introduction

It is well acknowledged that food is essential, but its vital role in the population’s health is often undervalued. However, this mindset has been slowly changing, as industry, researchers, and consumers are, nowadays, giving more attention to disease prevention and management through healthier diets, food safety, general well-being, and, more recently, the choice for more sustainable products and food production systems. The evident impact of food on human wellness has driven the urge to establish healthier dietary habits to fulfill food’s principal functions: supplying necessary nutrients, providing satisfaction, improving wellbeing, and regulating personal physiological states [1].

The search for food and food ingredients that provide additional benefits to consumers, called functional ingredients, that support or are supported by eco-friendly sustainable practices such as the reutilization of agro-industrial byproducts has been an area of increasing interest within the food industry and, recently, among consumers themselves [1]. Functional food additives are popular among consumers because of their improved organoleptic properties [2]. The development and commercialization of new functional ingredients to be incorporated in different food matrices now has a notorious impact on modern society’s dietary practices. In fact, ingredients such as probiotics and antioxidants are advertised as health promoters as a marketing strategy for food industries advocating the health benefits of their products (e.g., yogurts and fruit juices supplemented with probiotics and antioxidants, respectively).

The definition of functional food has been changing over the last few years; however, there is still no universally accepted single definition. Different international groups related to dietetics and nutrition (e.g., the International Food Information Council, the European Commission, and the American Dietetic Association) agreed that “functional food provides health benefits beyond basic nutrition” [1,3]. The international standards and guidelines for the evaluation of functional food are stipulated in the “Guidelines for use of nutrition and health claims (CAC/GL 23-1997)” of the Codex Alimentarius. When the established criteria are met, this codex allows the food sector to label its products with recognized health claims [4].

Functional foods are categorized into three classes based on their preparation: (1) conventional foods, (2) modified or fortified foods, and (3) food ingredients. Conventional foods are whole and unmodified foods (e.g., vegetables, meat, and cereal grains); fortified foods are regular foods supplemented with functional food components (e.g., calcium-fortified milk, anthocyanin-fortified bread, and vitamin-fortified honey); food ingredients are components from plants, microorganisms, and other inorganic raw materials that can be macronutrients, essential micronutrients, or non-nutrient components (e.g., inulin, *Lactobacillus*, iron) [1].

Overall, functional foods are meant to provide essential nutrients which can potentially bring additional health benefits to the host (e.g., stimulate the host’s immune system to prevent and control pathogenic infection) [3]. In the context of gut health promotion, prebiotics and probiotics are good examples of functional ingredients incorporated into food matrices. Prebiotics are substrates that are preferentially used by microorganisms within the host to provide health advantages to the host, whereas probiotics are live microbes that, when administrated in adequate quantities, deliver a health benefit to the host [5,6]. Numerous studies have helped us to understand the microorganisms’ functions and their beneficial impact on human health. Research with microorganisms, such as *Lactobacillus*, *Bifidobacterium*, *Saccharomyces,* and others, has made clear that their probiotic properties beneficially impact the gut’s health [7]. Other classes of food ingredients, such as polyphenols, polyunsaturated fatty acids (PUFAs), and phytochemicals have also been used to promote gut health [8].

Demands for more sustainable and eco-friendlier food production systems introduced the concepts of reduce, reuse, recover, and recycle, aiming towards a circular bioeconomy framework. Therefore, the use of byproducts originating from industrial processes to produce new ingredients to enhance staple foods can help diminish the environmental impact of food waste. Additionally, these byproducts are beneficial to health, reduce waste energy costs, and valorize underused functional ingredients as food supplements, thus improving the overall food system economy and sustainability [9,10].

This review focuses on defining effective strategies to identify potential new sustainable ingredients produced from agro-industrial byproducts and assesses their suitability for incorporation into food matrices as a primary tool to understand their impact on the host’s gut health.

## 2. Functional Ingredients and Sustainability

### 2.1. Fortified Foods

The increasing interest in new functional ingredients is related to the consumer’s demand for more natural, sustainable, and healthier foods in addition to concerns with self-care, aging, eco-friendly practices, and healthcare costs [1]. The food industry is aware of the consumer’s demands and interests, encouraging the development of new food products that meet those criteria. Table 1 shows some examples of foodstuffs supplemented with functional ingredients available on the market, which have beneficial effects related to the digestive, intestinal, immunological, bone, muscular, and/or nervous systems. Most of the incorporated functional ingredients are probiotics, dietary fibers, vitamins, or minerals.

The study of food matrices, with and without supplementation, and not only the isolated functional ingredients enables a better understanding of the ingredients’ real impact. The usage of functional ingredients must take into account the ingredient-food matrix interaction(s) as well as the possible negative consequences that may arise if they are handled wrongly [11]. It is important to consider the occurrence of these interactions, acknowledging not only the individual components’ potential but also the effects of the whole foodstuff once consumed and the benefits that it can effectively offer [12].

Food fortification is a common practice among the food industry worldwide with the priority of pursuing nutritional balance and promoting human well-being, especially in terms of gastrointestinal health and gut microbiota modulation. The high market potential of functional foods and nutraceuticals (i.e., any bioactive compounds of natural source that promote improvement on the host’s health and well-being) has turned them into multibillion-dollar industries, particularly when their positive health benefits, fewer or no side effects, and lower cost are touted in comparison to commonly used pharmaceuticals or drugs [13]. The global nutraceutical market (in which Europe, the USA, and Japan account for more than 90% of total market) was valued at USD 247 billion in 2019 with prospects to reach USD 336 billion in 2023, while a recent survey suggested that the global nutraceutical market may be valued in USD 340 billion in 2024 [14,15]. The development of new functional foods requires high research and development (R&D) costs and facing critical challenges involving the product development itself, regulatory issues, and public marketing of the developed products.

A deep understanding of the interactions of fortified foods is required. Emerging disciplines such as foodomics (the discipline that studies food and nutrition through the application of multiple advanced omics technologies), nutrigenomics (the discipline that uses molecular tools to assess and understand different responses gathered through a certain diet), and biotechnology (technology that works with cellular and molecular biological processes to develop products and/or techniques that may improve the quality of life) are crucial to support product development research [1,16,17,18]. With all these benefits in mind, a healthy lifestyle can be related to a balanced diet, capable of providing energy, nutrition, and stable gut microbiota activity [19].

### 2.2. Food Fortification within a Circular Economy Framework

Sustainability is a concept that has gained significant relevance in our daily lives, especially since the beginning of the 21st century. The scientific community and food manufacturers have been focusing their efforts on developing new, healthier, and more sustainable products, developing processes and services, and utilizing resources and/or goods that may answer basic needs and increase the community’s life quality. This approach promotes reductions in the use of natural resources and toxic materials as well as in emissions of waste and pollutants, increasing the safety of all participants in the system (workers and consumers) without compromising the needs of future generations [20,21]. Although there is no legal definition for sustainable foods and food ingredients, they are generally accepted as types of food obtained in a way that minimizes their negative impact on the environment (e.g., greenhouse gas emissions and carbon footprint), industry (e.g., production costs), and the population (e.g., healthier products) [21]. Today´s bioeconomy encompasses such ideals, aiming for zero environmental effect or mitigating climate changes and providing access to sustainable nutritive foods.

The agri-food industry generates a large amount of waste worldwide, with a tendency to increase, due to industrialization and urbanization [22]. This high production of food residues is a major environmental problem. The increase in agro-industrial activities worldwide (e.g., dairy, cereal, fruit, beer, and oil industries) and their inadequate byproduct/residue disposal (e.g., cereals and plants straws, husks, cereal bran, fruit pomace) have resulted in an increase in environmental problems such as soil, water, and air pollution, contributing to climate changes [9,10].

The Food and Agriculture Organization of the United Nations (FAO) estimates that one-third of the total food production for human consumption, i.e., approximately 1.3 billion tons of food, is lost and/or wasted annually worldwide [10,23]. One of the goals of the United Nations for 2030 is a reduction in food waste through strategies that decrease losses during production and in supply chains [23,24]. Although such loss cannot be avoided totally, a proposed mitigation strategy suggests resorting to biotechnology techniques for the recovery of bioactive compounds from agro-industry byproducts rather than treating these residues as worthless natural materials [9,24].

Despite the significant financial and scientific knowledge demands that the transition to sustainable food systems may require, there is no doubt that such a transition will tackle environmental issues and bring additional health benefits. This transition into a circular economy can be a way to re-utilize natural resources in an efficient and sustainable manner, decreasing the volume of unused or wasted biomass with high bioactive potential. The exploitation of these byproducts’ potential and their valorization and transformation into new functional ingredients can bring a new life to the fortified foods concept. What was formerly “waste” can then become value-added ingredients, contributing to consumer health, the circular economy, and sustainability [10].

A good example of an added-value agro-industrial residue is the lignocellulosic biomass resultant from cereal and plant harvesting. These residues, due to their chemical composition, are rich sources of bioactive compounds, such as dietary fibers, proteins, fatty acids, phenolic acids, and carotenoids, with a wide range of applications in the food, pharmaceutical, and/or cosmetic industries [10,21].

Food fortification with such bioactive materials can be a vital strategy against malnutrition that reduces agro-industrial waste [25]. However, a challenge for such food fortification processes is the product’s physicochemical and sensory properties. Possible interactions between the bioactive ingredient and the supplemented food matrix can change the final product´s properties. Consumer perception, labeling legislation, certification, definition of incorporation levels, formulations, potential to promote beneficial bioactivities, sensory qualities, and environmental impact are relevant questions that must be addressed [9,22]. As shown in Table 2, many approaches and new techniques are used in the search for bioactive compounds derived from agro-industrial byproducts to be used as supplements to improve foodstuff bioactive potential and functionalities for human health promotion [20,21]. Although plant-based byproducts are gaining more attention due to concern for animal welfare and changing dietary habits, a significant variety of animal-based bioactive components from fish (e.g., omega-3) and dairy industries (e.g., whey protein) are still used to supplement food products. Most of the existing studies focus on the physicochemical, microbiological, and sensory properties of the fortified final products, but few studies assess the bioaccessibility and bioavailability of such products in the human gastrointestinal tract (GIT) while even fewer studies assess the impact of these products on the human gut microbiota.

It is well acknowledged that gut microbiota plays a key role in the host’s health. Gastrointestinal health and gut microbiota modulation became a priority in managing and preventing a variety of metabolic diseases because modifying the metabolic activity of the intestinal gut microbiota may directly promote health [22]. Understanding food properties and recognizing ingredients’ impact on the host within different formulations are vital in assessing the impact that fortified foods may have on the consumer.

## 3. Gut Microbiota—A Perspective on Fortified Food Properties

Nowadays, it is generally accepted that microbiota refers to the entire collection of microorganisms (e.g., bacteria, archaea, viruses, fungi) existing in a specific location, while microbiome refers to the collection of all genetic material within the microbiota.

Gut microbiota includes all populations of beneficial (symbiotic) and/or harmful (pathogenic) microorganisms inhabiting the host’s gut, and the microbiome is the genome of all these microorganisms [52]. The microbial populations living in human gut microbiota are diverse (e.g., bacteria, archaea, and eukaryotes), abundant (from 10^10^ to 10^12^ live microorganisms per gram in the colon), and in a close relationship with the host [52,53]. Bacillota (formerly Firmicutes) and Bacteroidota (formerly Bacteroidetes) are the main phyla, followed by Pseudomonadota (formerly Proteobacteria) and Actinomycetota (formerly Actinomycetes). These four phyla represent 93.5–98% of the bacteria in the gut microbiota, and common genera include *Bifidobacterium*, *Lactobacillus*, *Bacteroides*, *Clostridium*, and *Escherichia* [54,55,56,57].

The progress of human gut microbiota studies enabled the understanding that everyone’s microbiota is unique and develops from early childhood to adulthood. In adulthood, it is relatively stable and does not go through significant changes; however, it is still susceptible to change [58]. The gut microbiota is important for the host’s gut health and general well-being, as it has functions related to gut development, mucosal immunity and other immune system interactions, food digestion, nutrient absorption, body detoxification, and the production of important metabolites such as short-chain fatty acids (SCFA), arginine, glutamine, vitamin K, and folic acid [59,60]. The microbiota composition is influenced by genetic and environmental factors. Environmental factors are extrinsic to individuals and can be controlled or changed during the life of individuals. Examples of these include geographical localization, toxin/carcinogen exposure, bacterial infections, antibiotic treatment, lifestyle, surgery, and diet [58].

Several research papers argue that the nutritional value of foods is partially affected by the composition of the individual’s gut microbiota and that, in turn, food shapes the composition of the microbiota [58,61,62]. It is currently unknown what classifies a “healthy” microbiota, but it is acknowledged that about 30 to 40% of the adult human gut microbiota can be modified during its lifetime and the factor with the greatest impact, accounting for more than 50% of such variations, is diet [58,61]. Daily, humans consume foods that enhance the activity of indigenous bacteria of the gut microbiota such as fermented milk, processed cheeses, and yogurts that support the delivery of probiotics to the GIT [62].

Dysbiosis and eubiosis are concepts related to gut microbiota health that are still not fully understood and not clearly defined, although in recent years, researchers have agreed on the definitions of these concepts [63,64]. Dysbiosis is a term used to describe an unhealthy state of the gut microbiota, referring to any variation in the populations and functions of the resident commensal microorganisms in comparison with the populations of resident commensal microorganisms observed in healthy people [63,64,65]. Dysbiosis can be caused by stress, medical interventions, diet, and external factors [64]. Eubiosis is a term used to describe the opposite of dysbiosis, that is, a “balanced” microbiota found in healthy individuals [63]. These two concepts are quite vague and possess little scientific value since the composition of microorganisms inherent to the gut microbiota of healthy individuals is highly variable. Even if all species of microorganisms and their genes were cataloged, it would not illustrate how a healthy microbiota community is [63,65].

Although dysbiosis is also an imprecise term, different types of dysbiosis have been distinguished: (a) growth of pathobionts (commensal microorganisms that have the capacity to cause pathology), (b) loss of biodiversity, (c) loss of commensal microorganisms, (d) loss of beneficial microorganisms, and (e) shifts in microbiota metabolic capacity [63,65].

Currently, the food, nutrition, and pharmaceutical industries struggle to translate the results of experiments related to the human microbiota, mainly due to the absence of a clear scientific definition of what a “healthy” microbiota is [63]. Probiotics and prebiotics are good examples of concepts with clear definitions nowadays; however, these definitions emerged only after being discussed for years among the scientific community [5,6]. Thus, the absence of a clear definition demonstrates a lack of scientific rigor in this research area, which can hamper progress, making it difficult to discuss the concepts, disseminate information into general society, and apply that information to industry [63,64,65].

There is an urgent need and interest for reproducible, reliable, cost-beneficial, and fast laboratory techniques to assess and describe the state of the microbiota before and after food intake. Different types of foods differently impact the gut microbiota, and there are diets that may be beneficial or harmful to the microbiota, consequently affecting the host’s health. Examples of these unhealthy practices are the consumption of low-fiber, high-fat diets (Western diet) containing processed foods, while diets rich in fiber are usually related to a healthier state [64].

Advances in gut microbiota research are often dependent on the development of new techniques, technologies, and methodologies. At the moment, several techniques and technologies are used in the study of probiotics and gut microorganisms which enable research into microbial genes, transcripts, and proteins, providing information that be applied in food science studies [17]. There has been an increase in articles that highlight the importance of the intestinal microbiota and its fundamental role in the development of innovative strategies for the prevention and treatment of human health conditions such as obesity, gastrointestinal illnesses, inflammatory conditions, and psychiatric disorders, among others [64]. In the last ten years, medical research estimated an investment of USD 1.7 billion in the human microbiome research field, which augmented the “microbiome market” and the private investment of companies/start-ups in a large scope of food, pharmaceutical, and cosmetic products [52].

The modulation of the gut microbiota through a specific ingredient or a fortified food is then of the greatest importance for the evaluation of the true impact that each fortified formulation may have on consumers. Studies at the gut microbiota level allow the evaluation and prediction of several possible outcomes when adding functional ingredients to food matrices.

Methodology related to the study of functional ingredients derived from agro-industrial byproducts and their impact on the gut microbiota is crucial, as most studies focus on the sensory and physicochemical properties of supplemented foods rather than the bioactivity potential of enhanced food matrices. The interaction of these ingredients with the gut microbiota can highlight the possible benefits to health from their supplementation to foods.

## 4. Methods Available to Evaluate Human Gut Microbiota Modulation

Nowadays, studies related to the human gut microbiome are one of the most dynamic research fields of science. The gut microbiome is a hot topic in both public and academic discussions of human health, considered by some to be our “last organ”, and we are slowly beginning to understand its importance as a promising area for new treatments [52,63,66].

The use of different omics techniques (e.g., culturomics, metagenomics, transcriptomics, metatranscriptomics, proteomics, metabolomics) in the microbiology field have allowed to identify microorganisms and their metabolites, expanding our understanding of the impact that these microorganisms and their metabolites have on nutrition and health [16,17,52]. Examples of techniques used to study gut microbiome/microbiota are described in Table 3.

The current challenges facing the food industry in the search for new functional ingredients to be added to food matrices can be overcome by combining individual omics techniques and obtaining basic insights regarding the effects of different compounds [17]. The common methodology, considered the “gold standard” for verifying bacterial viability in the food safety field, relies on culture-dependent methods [67,68]. However, culture-dependent methodology has its limitations, such as an inability to quantify uncultured bacteria, which represent around 60 to 80% of the bacterial populations present in gut microbiota [66,69]. The development of metagenomics techniques, such as quantitative polymerase chain reactions (qPCR), assists in overcoming these limitations, providing useful tools for the identification, description, and understanding of the role of each bacterial group in the microbiota [66,70]. But even these metagenomics techniques have their own limitations, as reviewed by Fraher et al. in 2012 and Shang et al. in 2018 [71,72]. Therefore, culturomics and metagenomics can be considered as techniques that complement each other and that can overcome each other’s limitations, depending on the proposed experimental design and the objective of the work.

One of the omics that is extensively used in food science is metabolomics, which is a set of techniques for studying the metabolic pathways of biological systems by detecting and quantifying the production and/or changes of metabolites stimulated over time by biological systems (e.g., microbial communities) [17,73]. The food industry has a keen interest in analyzing the potential impacts of foods and food additives on metabolites produced by the gut microbiota, such as SCFA and especially butyrate, as these metabolites have a direct impact on the psychological state of the host [17]. Metabolomics analysis comprises two main approaches: directed analysis (determination and quantification of a group of pre-defined metabolites) and metabolic profiling (detection of all metabolites and/or their products) using a particular technical analysis accompanied by an absolute or relative quantity estimative and relying on chromatography techniques such as high-performance liquid chromatography (HPLC) and gas chromatography (GC) to separate compounds to be identified and quantified [16,73].

In most of the human microbiota (models or microbial mixed communities) impact studies, a combination of metagenomics and metabolomics techniques are used, with some studies using culture-dependent techniques to complement their experiments and collect valuable data to determine the impacts of specific compounds on microbial communities [16,66,74].

**Table 3 foods-12-02209-t003:** Example of techniques used in methodologies to study the gut microbiome/microbiota.

Technique	Description	Function	-Omic	References
Culture	Isolation of bacteria on selective media	To quantify culturable viable bacteria present in biological samples	Culturomics	[66,71]
Quantitative polymerase chain reaction (qPCR)	Amplification and quantification of 16S rRNA. Reaction mixture contains a compound that fluoresces after binding to double-stranded DNA	To identify and quantify the presence of a specific microorganism in biological samples	Metagenomics	[71,75,76]
Denaturing or temperature gradient gel electrophoresis (DGGE)/(TGGE)	Chemical or temperature denaturation and gel separation of 16 rRNA amplicons	To characterize microbial communities and their functional genes in biological samples		
Fluorescence in situ hybridization (FISH)	Hybridization of fluorescent labeled oligonucleotide probes with target 16S rRNA complementary sequences. This approach can be coupled with a special microscope or to flow cytometry to enumerate the number of fluorescence events	To identify and quantify the presence of specific live microorganisms in biological samples		
Microbiome shotgun sequencing	Random break-up of the whole genome into small DNA fragments followed by parallel sequencing of each fragment. A computer program analyzes the results of the DNA sequences to reconstitute the whole genome.	To determine the DNA sequences of the whole genome in the biological samples. To characterize, identify, and quantify the microbial communities present in the biological sample		
High-performance liquid chromatography (HPLC)	Chemical separation of components in a liquid mixture. The liquid sample is injected into a pressurized liquid solvent (mobile phase) that goes through a column packed with a separation medium (stationary phase). Each component present in the sample interacts with the stationary phase, separating by a process of differential migration during the time spent travelling through the column. This process is monitored by a computerized system of detectors.	To identify and quantify specific metabolites present in biological samples (e.g., SCFAs)	Metabolomics	[77,78]
Gas chromatography (GC)	Chemical separation of components in a liquid or gaseous mixture. The liquid or gaseous sample is injected into a carrier gas (mobile phase) that goes through a column (stationary phase). The column is inside of an oven that regulates the temperature of the carrier gas and the eluent that leaves the column. This process is monitored by a computerized system of detectors.			

## 5. Proposed Strategy to Assess the Impact of Fortified Foods on the Gut Microbiota

The assessment of the potential impacts of functional food ingredients incorporated in food matrices on the human microbiota has been a “knowledge platform” for the industry to inspire the creation of new products that reach global markets. The industry takes advantage of this “knowledge platform”, as several *in vitro* and *in vivo* studies (with animals), as well as clinical trials (with human volunteers) (recommended following this order) are required before these ingredients are commercially available.

As a first-stage strategy to screen functional ingredients, the utilization of *in vitro* simulation models is considered the most suitable approach, since it is less laborious and time-consuming, reduces the use of *in vivo* models, enables simulation of the different conditions occurring in the gut, and raises fewer ethical problems than animal *in vivo* studies and human clinical trials [79,80]. These *in vitro* models can be used to screen the product’s nutritional value and its digestion process and to evaluate their effects on microbial populations, such as the inhibition of pathogenic bacteria growth [81,82]. Although *in vitro* techniques are important, human *in vivo* clinical trials are still needed to thoroughly investigate the “true” impact that a supplemented food matrix containing a specific food component may have in humans.

The strategy proposed by the authors is to assess the potential of new and sustainable food ingredients and the possibility of their incorporation into different food matrices, understanding their effects on gut microbiota modulation, in pursuit of the best option to incorporate functional ingredients for the fortification of food matrices. This approach encompasses (1) fermentability preliminary assessments, followed by (2) simulations of gastrointestinal digestion and gut fermentation and the identification/quantification of bacteria and their metabolites (resorting to microbiology, biochemistry, metagenomics, and metabolomics techniques), and, finally, (3) human clinical trials (Figure 1). This approach is in agreement with the study published by Scott et al. 2020, who explored the same idea for the evaluation of potential prebiotics [83].

### 5.1. Stage 1—Fermentabilty Assay

The aim of this step is to understand the impact of a food ingredient in a specific inoculum according to the desired purpose and evaluate if the ingredient or food product is fermentable. The reason behind this approach is to screen microbial interactions and cell functions since microbial metabolism can provide information regarding microbiology systems [16]. With this approach, it is possible to assess different parameters, such as bacteria growth, organic acid production, pH, and ammonia (NH_4_^+^) production, enabling the evaluation of the saccharolytic and proteolytic activity of the inoculum.

Some studies related to the gut microbiota focus on the impact that ingredients have on specific bacterial species such as *Lactobacillus*, *Bifidobacterium*, *Faecalibacterium prausnitzii*, *Escherichia coli*, and *Salmonella* spp. These studies usually start by assessing monocultures of specific bacteria and progress to mixing one monoculture with another bacterial monoculture (co-cultures) or with more bacterial species (consortium). These types of bacterial species-specific studies enable us to see the interactions between probiotics and pathogens or probiotics and probiotics within the substrate being tested. An example of a study that carried out this type of monoculture and bacterial consortium assay is the one developed by Carvalho et al. (2019), which studied the impact of insect flour on bacterial monocultures and a consortium of *Lactobacillus* and *Bifidobacterium* (two of the most important genera of gut bacteria for their probiotic activities), testing two different percentages, i.e., 1 and 10% of each monoculture [84].

Nowadays, the use of microorganisms belonging to a specific ecosystem to study biological systems is considered practical and useful. The utilization of stool samples to assess the composition and functionality of the human gut microbiota is an example of a biospecimen used to study biological systems. It is common practice to use fecal samples as a proxy of the gut microbiota due to the convenient, non-invasive, feasible, and large volume of sample collection; however, scholars recognize that fecal microbial populations do not fully represent whole gut microbiota and that feces are more representative of luminal microorganisms than the mucosa-associated microorganisms of the GIT [85,86].

Nevertheless, stool samples guarantee a good representation of bacterial communities present in the lumen of the host’s colon, since a significant portion (55–60%) of the biomass consists of bacteria [87,88]. Depending on the study being carried out, it is possible to collect fecal inoculum representing the gut microbiota of healthy people or people suffering from specific chronic diseases (e.g., rheumatoid arthritis, Crohn’s disease, colon cancer, irritable bowel disease) and assess the effect that certain food products have in their bacterial populations. The physicochemical properties of the inoculum are one of the major concerns in different types of experiments (e.g., *in vitro* assays), as different factors affect the microbial activity. Factors such as subjects and their diet, sampling time/day, storage conditions, inoculum preparation procedures, and concentrations used will directly influence the microbial composition, viability, and activity of the ecosystem that is simulated [89,90,91].

Depending on the purpose of the research, different types of inoculums (e.g., monocultures, co-cultures, consortium, fecal inoculum) can be used to assess the impact of an ingredient (Figure 2).

To fully understand the fermentability assay, a schematic protocol is shown below as an example (Figure 3); nevertheless, other experimental designs can be used [84,92,93,94].

Briefly, sterile batch culture fermentation-independent tubes are set up and aseptically filled with sterile basal nutrient medium (BNM) [75] (in triplicates for each sampling time) with the pH adjusted to around 7 and gassed overnight with O_2_-free N_2_ inside of an anaerobiotic workstation at 37 °C. Each tube is inoculated with a specific inoculum (e.g., 10% (*w*/*v*) fecal inoculum diluted in 0.1 M phosphate-buffered saline (PBS)). The pH value is monitored at specific times (e.g., 0, 24 and 48 h). The batch tube cultures fermentations are performed under anaerobic conditions at 37 °C at a specific time (e.g., 48 h), during which samples for SCFA, branched chain fatty acid (BCFA), and lactate analysis are collected and analyzed through metabolomic techniques (e.g., HPLC) and bacterial enumeration is conducted by culture-dependent (e.g., plating) or culture-independent (e.g., qPCR) techniques [95]. After collecting the samples, a culture-dependent technique can be performed or the samples can be centrifuged, with the supernatant collected for metabolomic techniques and the pellet collected for metagenomic techniques (Figure 3).

### 5.2. Stage 2—Human GIT Simulation Model

With a focus on maximizing their beneficial potential, the impacts of the different possible combinations of functional ingredients and selected food matrices can be determined, taking into account all the events and interactions along the GIT and the impact on the host´s gut microbiota modulation [67,83]. According to the purpose and circumstances, different gastrointestinal simulation models and approaches can be used, such as SHIME (Simulator of Human Intestinal Microbial Ecosystem), PolyFermS (Polyfermentor Intestinal Model), or MIMICS (Model Intestinal Microflora in Computer Simulation), among others, as described by Dixit et al. (2023) and Isenring et al. (2023) [96,97]. In the present review, the authors illustrate one reliable and reproducible approach to human GIT modelling that they have used and applied to their research.

The *in vitro* gastrointestinal protocols are capable of mimicking the physicochemical and physiological conditions of the GIT. This is highly relevant, as they enable the study of digestion and absorption, as well as the assessment of nutrient bioavailability and bioaccessibility during digestion. This process can be regarded as three sequential steps. Step one considers a simulation protocol of the gastro-intestinal digestion, step two considers the mimicking of intestinal absorption, and step three considers the performance of the colonic fermentation.

Considering step one, different gastrointestinal simulation protocols can be found in the literature [98,99,100,101]; nevertheless, the authors recommend the use of a well-known and consolidated protocol—INFOGEST 2.0—which is a standardized European protocol that simulates oral, gastric, and intestinal digestion, taking important factors, such as enzymes, temperature, and pH, into account [101]. This protocol has been used for several food matrices (e.g., bread, infant formula, milk, cheese) and optimized differently by several research groups according to the aim of their work [102,103,104]. Step two mimics the intestinal absorption of the digested samples. Finally, step three concerns the colonic fermentation of the unabsorbed digesta from step two [76,77].

To better understand the proposed strategy for Stage 2, the authors provide a brief explanation of how to proceed with this protocol, referring to a selected study case developed by the authors themselves.

Before step one, a preparatory step is necessary, consisting of selecting the target food matrix on which the selected functional ingredient will be incorporated to assess its suitability, e.g., whether it is beneficial, neutral, or harmful. The most common food matrices to be fortified with functional ingredients such as probiotics and prebiotics are dairy products; however, there is increasing interest in incorporating functional ingredients into non-animal and/or non-dairy food matrices [62,105,106,107]. In the scientific literature, the supplementation of functional ingredients in food/feed matrices for both humans and animals is between 0.5–10%, with most ingredients being tested in a range of 1 to 3% [105,108,109,110,111].

Following these trends, the authors, selected skim milk (SKM) as food matrix and tested three conditions (in duplicate): (1) SKM without supplementation, (2) SKM supplemented with 1% fructo-oligosaccharides (FOS), and (3) SKM supplemented with 1% *Saccharomyces boulardii* (SB).

A simulation protocol described by Brodkorb et al. (2019) [101] is then used to simulate the gastrointestinal digestion protocol (step one), followed by an intestinal absorption protocol (step two), and colonic fermentation protocol (step three), as previously described in de Carvalho et al. (2021) and de Carvalho et al. (2022) [76,77] (Figure 4a).

Additionally, a inoculum control (IC) condition (in duplicate) was also included in the step three. This condition refers to the fecal inoculum without any source of nutrients. Batch fermentations were run at 37 °C for 48 h and, during the fermentation run at different time points (e.g., 0, 24, and 48 h), similar parameters to those of the fermentability assay could be evaluated according to the sample procedure shown in Figure 4b (e.g., bacterial, organic acid, and NH_4_^+^ quantification). This fermentation procedure is analogous to several previously reported fermentation protocols; however, changes in basal media, proportion of inoculum, and pH control throughout fermentation can be discovered [94].

This stage (Stage 2—Human GIT simulation model) will also enable comprehension of how the incorporated functional ingredients (in a specific food matrix) can impact the growth of gut microbiota beneficial bacteria such as *Bifidobacterium* and *Lactobacillus*. These bacteria, along with their produced metabolites, can then be used as health indicators and as references for targeted nutrition strategies [112].

### 5.3. Stage 3—Human Clinical Trials

Human clinical trials are the last step to validate all the *in vitro* results obtained previously. For those who are interested in carrying out such studies and intend to implement human nutrition randomized controlled trials (RCT), recently, a series of manuscripts describing the best practices for designing, conducting, documenting, and reporting these types of studies were published in the Advances in Nutrition journal [113,114,115,116,117]. Additional guidelines can also be found in the CONSORT (Consolidated Standards of Reporting Trials), created to improve the quality of clinical trial reports [118].

Human trials must be examined and authorized scientifically and ethically according to internal regulations, legally established based on research ethics guidelines, and approved by the scientific and ethical commission where the trials will be performed. An informed consent form within regulations, clarifying the purpose of the study, and accepted by the ethical commission must be signed by subjects involved in the study, with consent given to collect and process/use their personal information, health parameters, and biological samples. Additionally, power analyzes are required to determine the minimum number of subjects (n) for the trial. Power analysis is composed of four components: (1) statistical power, (2) sample size, (3) significance level (α), and (4) expected effect size [119]. Most studies use an α of 5%, 80–95% power, and generally, an effect size based on similar studies or a study pilot [119,120,121,122]. Currently, there are software programs (e.g., G*power) that allow the calculation of the minimum sample size for the trial (necessary to stipulate effect size, α, and power).

Indeed, RCTs are regarded as the most reliable data on the efficacy of interventions, since they minimize the risk of confounding by other factors, allowing for the establishment of a causal relationship between the interventions and health outcomes evaluated [118]. However, this type of human trial is usually difficult to carry out in a complete way due to the short duration of the trial and lack of resources, namely people and subjects. In such cases, other methodological approaches are found so human trials can be carried out within the study objectives.

An example of a non-RCT human trial is when two different groups of volunteers (i.e., a control group vs. test group) are used, to study the impact of daily consumption of a specific ingredient for a certain amount of time, usually one month, on the increase in selected bacteria from the gut microbiota (e.g., *Bifidobacterium*).

First, the experimental design states the aim of the experiment, the null and alternative hypotheses, and then the methodology foreseen is explicitly defined. The questions asked in the forms need to be structured in a way that participants can provide their personal information within the scope of the study and give their signed consent. Then, the effect size of trials is researched to define the significant level and power intended in the result analysis (Figure 5). The next step is to submit the work plan to the ethics and scientific council of the institution where the trials will be performed and then, only after the study’s approval, the volunteers’ recruitment can start. In the hypothetical scenario, in which only 40 volunteers were recruited, the effect size for the experiment is 1–1.5. By setting the α and power to 5 and 95%, respectively, and using the G*power software, each group should have 11 to 23 volunteers. Due to this number of volunteers (40), volunteers will be divided into two groups—(a) those who will not consume the testing ingredient (control group) and (b) those who will consume the testing ingredient (test group). To make the n of the two trial groups equal, it was determined that the control and test groups would have 20 volunteers each. The volunteers from each group will go through two different testing periods: (1) for the two first weeks (period of adaptation to the food matrix) both groups will consume the food matrix without the target functional ingredient and their stools will be analyzed weekly. Additionally, bowel habits, sleep, satiety, and vital signs, among other parameters, can be monitored. After that period, (2) the test group start to consume 1 teaspoon of the target ingredient (2.5–4 g) with milk daily and for four weeks (period of consumption of the tested ingredient) while the control group continues to consume milk without any added ingredient for the same period. The same analyses should be performed on a weekly basis to determine the differences between the two periods. The results obtained during the complete trial will be analyzed to determine whether, in fact, the daily consumption of milk with the target ingredient, over a month, had an effect on the volunteer’s gut microbiota (Figure 5).

## 6. Conclusions

It is well acknowledged that food fortification is of extreme relevance for nutritional purposes and health maintenance, and in such considerations, the concepts of sustainable processes and circular economy need to be taken into account. However, it is necessary to deeply understand what the effects of each target functional ingredient on the human organism are, with special attention to the gut microbiota, and if such effects can vary when different food matrices are used. This review demonstrates that there are multiple approaches to assess the impact of food ingredient supplementation on gut microbiota modulation and that, with the proper methodologies, specific goals can be better accomplished. One strategy is illustrated in this review as an example of how several methodologies can be used to obtain valuable data. New perspectives then can arise, leading to the development of new fortified foods, not only with benefits to the host but also encompassing today´s challenges of sustainability.

From a circular bioeconomy point of view, the urgent demand for the use of agro-industrial byproducts as sources for new functional/bioactive food ingredients is a key action to prevent waste and diminish the environmental impact of the agriculture-food industries. The incorporation of such ingredients should, at the same time, add value to existing foodstuffs with the aim of improving the host´s health, namely through gut modulation. Before entering the market, supplemented products should be well-tested to understand if the stated effect or health claim still exists once the food matrix is consumed. This can enable the food industry to improve food formulations or develop new functional ingredients according to the targeted food matrices, responding to consumers’ demands and encouraging healthier and sustainable dietary habits.

Therefore, encouragement of the application of microbiota assessment methodology to the food industry can be a turning point, helping to increase the number of fortified staple food matrices with sustainable bioactive ingredients and to find new sustainable functional ingredients to be incorporated in our diets.

## Figures and Tables

**Figure 1 foods-12-02209-f001:**
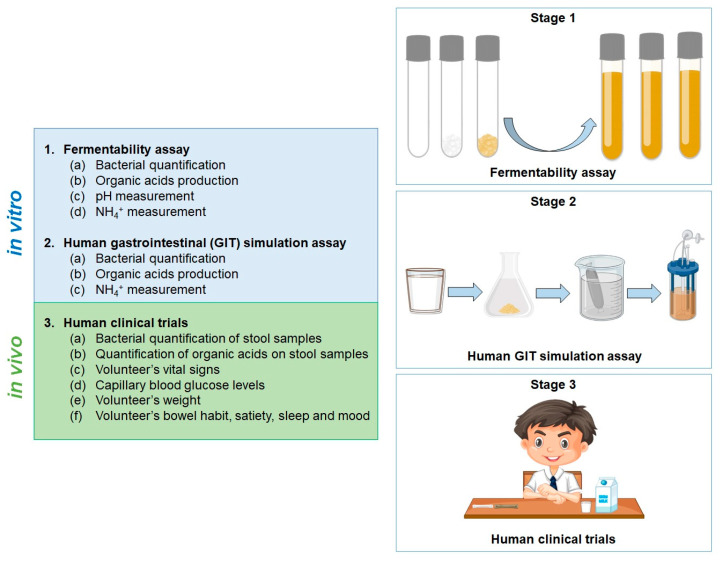
Strategy proposed by the authors to assess the potential of new and sustainable functional food ingredients when incorporated into a food matrix.

**Figure 2 foods-12-02209-f002:**
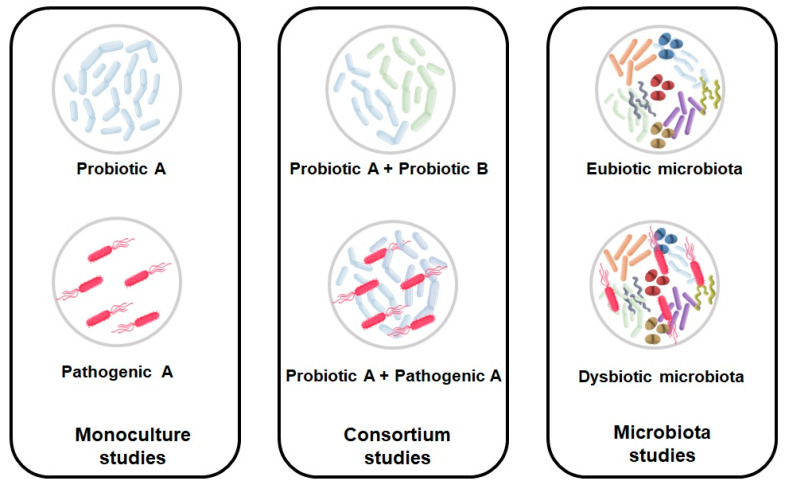
Different types of inoculums that can be used in fermentability assays.

**Figure 3 foods-12-02209-f003:**
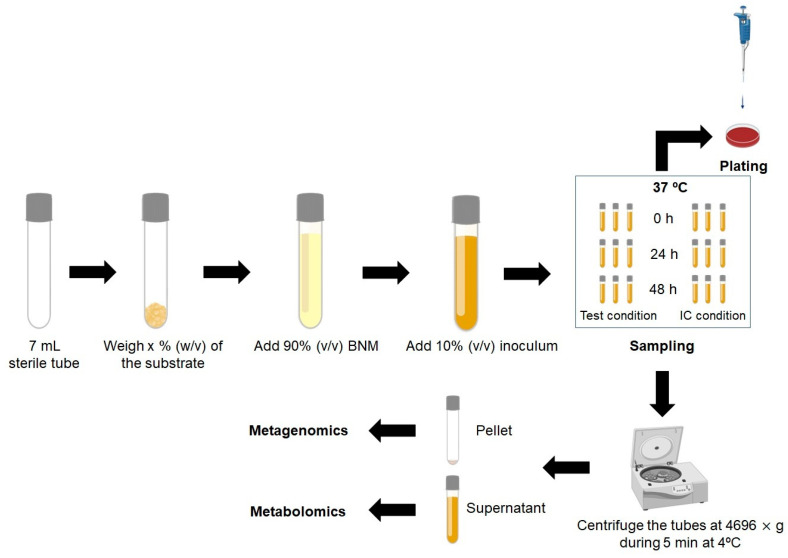
Stage 1, fermentability assay protocol suggested by the authors. BNM—Basal nutrient medium; IC—Inoculum control.

**Figure 4 foods-12-02209-f004:**
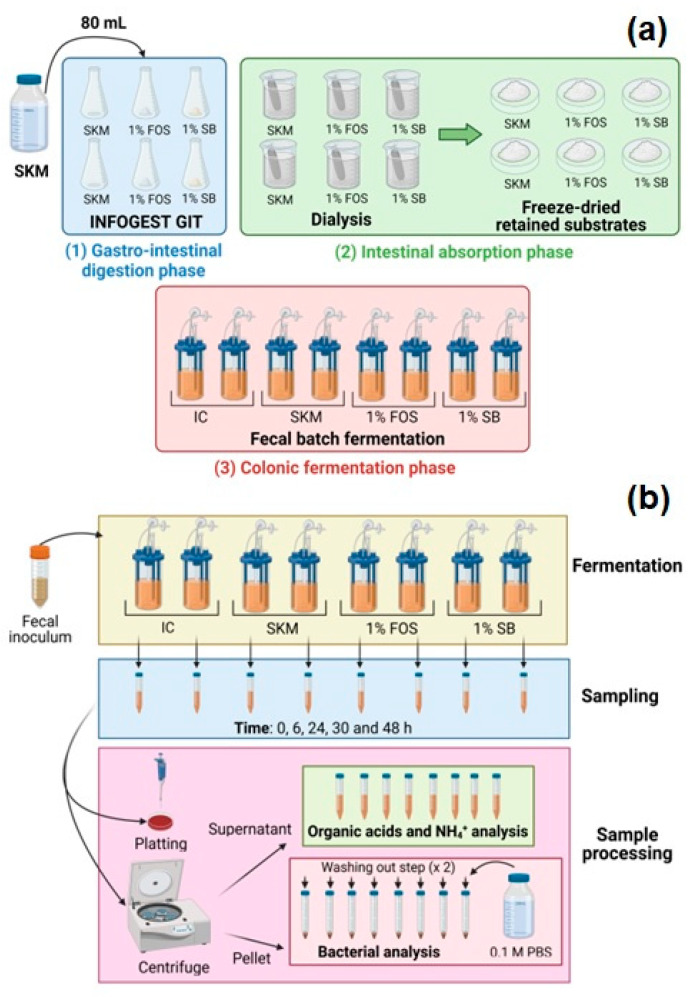
Stage 2, (**a**) the human gastrointestinal tract (GIT) protocol and (**b**) sample processing suggested by the authors. FOS- Fructo-oligosaccharides; IC- Inoculum control; SKM- Skim milk; SB- *Saccharomyces boulardii*. The figure was created with BioRender.com.

**Figure 5 foods-12-02209-f005:**
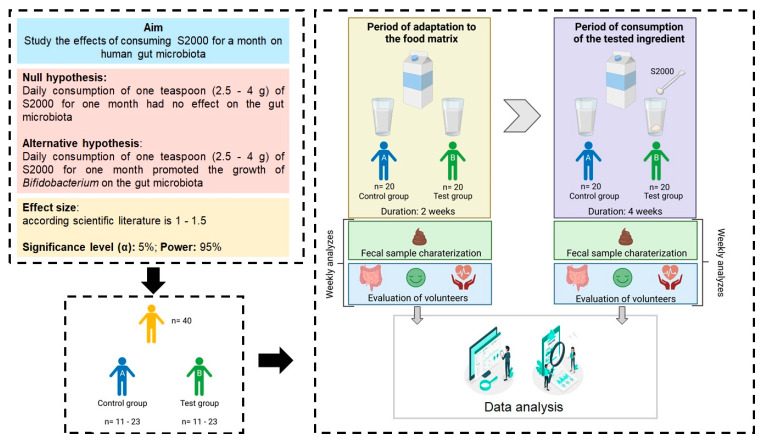
Stage 3, example of a non RTC human clinical trial suggested by the authors.

**Table 1 foods-12-02209-t001:** Examples of fortified foods commercially available.

Product Name	Food Matrix	Food Ingredient Incorporated	Claim
Blevit Plus 8 Cereals and Cookie Maria	Cereal baby porridge	Fructo-oligosaccharides (FOS), *Bifidobacterium infantis*, *Lactobacillus rhamnosus* and vitamin complex (e.g., vitamin A, C, and D)	Provides essential micronutrients, encourages normal bone growth and development and maturation of the baby’s digestive and immune systems
Website: https://www.blevit.com/producto/blevit-plus-duplo-8-cereales-y-galletas-maria (accessed on 25 May 2023)			
Danone Activia	Yogurt or fermented skimmed milk	*Bifidobacterium animalis* CNCM I-2494	Reduces the frequency of intestinal discomfort
Website: https://www.danone.pt/marcas/activia (accessed on 25 May 2023)			
John West Energy tuna steak	Tuna	Vitamin B	Reduces tiredness and fatigue
John West Immunity tuna steak		Vitamin C	Supports immune system
John West Heart tuna steak		Omega-3	Supports heart function
Website: https://www.john-west.ie/products/range/nutrient-rich-tuna/ (accessed on 25 May 2023)			
Marigold Vegan Engevita	Yeast flakes	Vitamin B12	Does not mention the possible outcomes the consumption of it brings, only mention what is included in the food matrices
Marigold Super Boost Vegan Engevita		Vitamin D and iron	
Website: http://marigoldhealthfoods.co.uk/products/engevita/ (accessed on 25 May 2023)			
Mimosa Bem Especial	Milk	Calcium and vitamin D	Supports the growth and development of bone mass
Website: https://mimosa.com.pt/produtos-lacteos/leite/bem-especial/calcio/ (accessed on 25 May 2023)			
Myvitamins wellness Gut gummies	Mixed berry flavor gummies	*Bacillus coagulans* and vitamin C	Improves health and well-being, supports immune system, and helps to reduce fatigue
Website: https://www.myprotein.com/vitamins/gut-gummies/12552274.html (accessed on 25 May 2023)			
Nestle Bolero	Cereal and fiber-soluble powder	Inulin	Restores energy and maintains the person’s well-being
Website: https://saboreiaavida.nestle.pt/produtos/cafe-e-bebidas/bolero-cereais-e-fibra (accessed on 25 May 2023)			
Nestle Kefir Natural	Pasteurized semi-skimmed milk	Kefir grains and yeast	Supports the digestive and immune systems
Website: https://saboreiaavida.nestle.pt/produtos/lacteos-e-sobremesas/kefir-natural-150g (accessed on 25 May 2023)			
Sonatural Culturas vivas	Apple juice, carrot ginger juice, or pineapple ginger juice	Inulin, *B. coagulans* GBI-30 6086, and vitamin C	Improves digestive health, stimulates the immune system, maintains equilibrium of gut microbiota and reduces the activity of harmful bacteria.
Website: https://sonatural.pt/collections/shotsprobioticos (accessed on 25 May 2023)			
Vatel Iodized Coarse Sea Salt	Sea salt	Iodine	Supports the normal production of thyroid hormones as well as thyroid function, the nervous system, and cognitive function
Website: http://vatel.pt/en/iodized-cooking-sea-salt-1kg/ (accessed on 25 May 2023)			
Yakult Original	Fermented skimmed milk	*Lactobacillus casei* Shirota	Helps to keep a balanced gut microbiota
Yakult Light		*L. casei* Shirota, vitamin D and E	Helps to keep a balanced gut microbiota, supports the immune system, maintains bone and muscle function, and protects cells from oxidative stress
Website: https://www.yakult.co.uk/products/ (accessed on 25 May 2023)			

**Table 2 foods-12-02209-t002:** Examples of different agro-industrial byproducts used to produce bioactive compounds to be incorporated in food matrices.

Food Matrix	Source (Byproducts) of Bioactive Compounds	Example of Bioactive Compounds Present	Incorporation Tested (%)	Type of Study	References
Bread	Mango	Carotenoids and polyphenols	5–25	Physicochemical studies	[26]
	Lettuce	Fibers and vitamins	2–40	Physicochemical and sensory studies	[27]
	Onion	Fibers and polyphenols	0.1–5	Physicochemical studies	[28,29]
Cakes	Grape	Fibers and polyphenols	15–25	Physicochemical and sensory studies	[30,31]
	Watermelon and melon	Carotenoids and vitamins	5–15		[32]
	Broccoli	Fibers and glucosinolates	2.5–7.5		[33]
Cheese	Mushroom	β-glucans	0.4	Physicochemical and sensory studies	[34]
	Fish	Essential fatty acids	1	Physicochemical and microbiological studies	[35]
	Asparagus	Anthocyanins and fibers	0.5 to 1.5	Physicochemical and sensory studies	[36]
Cookies	Apple	Fibers and polyphenols	10–20	Physicochemical, sensory, and /or *in vitro* studies	[37,38]
	Pomegranate		2.5–10		[39,40]
	Orange		5–20	Physicochemical and sensory studies	[41]
Fermented milk	Chestnut	Minerals and vitamins	2	Physicochemical and microbiological studies	[42]
	Passion fruit	Fibers and vitamins	1	Physicochemical, microbiological, and sensory studies	[43]
	Rice	Fibers and minerals	1–3		[44]
Meat	Tomato	Carotenoids and fibers	1.5–6	Physicochemical and sensory studies	[45]
	Pineapple	Fibers and vitamins	1.5		[46]
	Banana	Fibers and minerals	2–6		[47]
Yogurt	Carrots	Carotenoids and vitamins	2.5–20	Physicochemical, microbiological, and/or *in vitro* studies	[48,49]
	Dairy	Proteins and peptides	0.33–1	Physicochemical studies	[50]
	Coffee	Fibers and polyphenols	2–6	Physicochemical, microbiological, and *in vitro* studies	[51]

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
