# Peer review of "Strategies to Assess the Impact of Sustainable Functional Food Ingredients on Gut Microbiota"

_foods, 2023, doi:10.3390/foods12112209_

Round 1

Reviewer 1 Report

Reference 5 should cite the original ISAPP-authored papers defining prebiotics and probiotics.

Section 4 para 2: Firmicutes and Bacteroidetes are now also known as Bacillota and Becteroidota, respectively. Up to date usage may improve the contemporary feel of this manuscript. "The most common genera" would be better written as "Common genera include...". Please cite more significant papers than references 53, 54.

Figure 3 and associated methodological text is of limited value in a review article of this scope. Better to instead suggest that various tube fermentation methods are available and cite them than attempt to duplicate one, even as an example. In Figure 3 itself, on the top right of the image, the word should be "plating" not "platting".

Figure 4 image resolution appears poor. As with Figure 3, this figure and associated methodological text is of limited value in a review article of this scope. Better to instead suggest that other GIT models are available and cite them than attempt to duplicate one, even as an example.

Section 6.3 para 2 contains unnecessary detail about power analyses, out of keeping in a review article of this scope. Better to simply state that adequate power is required, and cite other papers for people to look for themselves.

Similarly, Section 6.3 paras 4 and 5 contain unnecessary details of a study design, out of keeping in a review article of this scope.Again, simply cite examples please.

Reviewer 2 Report

The authors did a great job by explicitly collating this review. It was well written, interesting and scientifically worthy of publication. A minor points for the authors to examine are listed below

·        Provide some reference links to information in Table 1

·        Add references to methodologies listed in Table 3

·        I would suggest a rearrangement of the sections e.g. 3 could be a sub-section of 2.

·        Are there limitations in this summary, and future perspectives should be added
